evolution

age structure, assortative mating, dosage compensation, intragenomic conflict, inclusive fitness, sex chromosome

**Author for correspondence:**
Thomas J. Hitchcock
e-mail: th76@st-andrews.ac.uk

# A gene's-eye view of sexual antagonism

Thomas J. Hitchcock and Andy Gardner

School of Biology, University of St Andrews, St Andrews KY16 9TH, UK

  TJH, 0000-0002-6378-5023; AG, 0000-0002-1304-3734

Females and males may face different selection pressures. Accordingly, alleles that confer a benefit for one sex often incur a cost for the other. Classic evolutionary theory holds that the X chromosome, whose sex-biased transmission sees it spending more time in females, should value females more than males, whereas autosomes, whose transmission is unbiased, should value both sexes equally. However, recent mathematical and empirical studies indicate that male-beneficial alleles may be more favoured by the X chromosome than by autosomes. Here we develop a gene's-eye-view approach that reconciles the classic view with these recent discordant results, by separating a gene's valuation of female versus male fitness from its ability to induce fitness effects in either sex. We use this framework to generate new comparative predictions for sexually antagonistic evolution in relation to dosage compensation, sex-specific mortality and assortative mating, revealing how molecular mechanisms, ecology and demography drive variation in masculinization versus feminization across the genome.

## 1. Introduction

New genomic approaches paint an increasingly vivid picture of the extent of sexual antagonism across the genome, identifying specific loci at which fixed or segregating alleles increase the fitness of their female carriers while decreasing the fitness of their male carriers, or *vice versa* [1,2]. The overall action of natural selection on such alleles depends on how the benefits enjoyed by one sex are balanced by costs incurred by the other, and since different parts of the genome are expected to place different values on the fitness of females and males this is predicted to lead to an intragenomic conflict of interest with respect to sexually antagonistic traits [3–8]. Conventionally, the X chromosome has been viewed as placing twice as much value on the fitness of females as it does the fitness of males, on account of it spending twice as much evolutionary time in the bodies of females than in the bodies of males, whereas the autosomes have been viewed as placing equal value on each sex on account of them spending an equal portion of evolutionary time being carried by males and females [9–16]. Accordingly, the X chromosome and the autosomes have been regarded as being locked in an intragenomic conflict, in which the former favours phenotypes that are relatively closer to the female optimum and the latter favour phenotypes that are relatively closer to the male optimum [7,8].

However, this view has been challenged by recent mathematical analysis which has indicated that male-beneficial alleles may be more—not less—readily favoured at X-linked loci than at autosomal loci [17,18]. Specifically, this work suggests that while the condition for an autosomal sexually antagonistic allele to invade from rarity is the same irrespective of which sex obtains the benefit, the condition for an X-linked sexually antagonistic allele to invade from rarity is almost always less stringent when males obtain the benefit and females suffer the cost than when females obtain the benefit and males suffer the cost, where benefits and costs are defined according to how the allele's homozygous and hemizygous genotype fitnesses differ from those of the resident allele. Empirical support for masculinized X chromosomes has been found in humans [19], aphids [20] and stalk-eyed flies [21]. These surprising results have been

interpreted as directly contradicting evolutionary biologists' classic understanding of intragenomic conflict [17].

Here, we show that these results are, in fact, fully consistent with the classic view, by taking an explicit gene's-eye-view approach that considers the inclusive-fitness interests of a single gene rather than a whole genotype [8,22]. By partitioning a gene's 'agenda' (valuation of female versus male fitness) from its 'power' (ability to exert fitness effects upon females versus males), we show that the classic view concerns a gene's agenda and the discordant results emerge from sex differences in power. We use this framework to generate new comparative predictions for sexually antagonistic evolution in relation to dosage compensation, sex-specific mortality and assortative mating, revealing how molecular mechanisms, ecology and demography drive variation in masculinization and feminization across the genome.

## 2. Results

We begin by recapping the puzzling mathematical results that have motivated our analysis. Traditionally, X-linked genes, for which there is a double-dose in females in comparison with males, have been viewed as placing twice as great a value upon the fitness of females as that of males, on account of their spending twice as much evolutionary time in the bodies of females as opposed to males [8,23–26]. However, specific population-genetic models of sexual antagonism have cast doubt on this principle. If a mutant allele confers a fitness benefit $S$ to one sex and confers a fitness cost $T$ to the other sex when in its homozygous/hemizygous form, then in the absence of dominance effects the condition for natural selection to favour invasion of the allele from rarity turns out to be $S > T$ for both X-linked and autosomal genes, irrespective of which sex obtains the benefit [15,17]. That is, the X chromosome does not appear to be particularly biased towards female-beneficial alleles versus their male-beneficial counterparts.

The situation is more complex in the presence of dominance effects. Rice [15] showed that whereas the condition for a sexually antagonistic allele to invade from rarity on an autosome remains $S > T$, the corresponding condition for the X chromosome is $S > 2hT$ if it is male-beneficial and $S > T/(2h)$ if it is female-beneficial, where $h$ is the dominance coefficient. Accordingly, if the degree of dominance is the same for both male-beneficial and female-beneficial alleles, then the X chromosome is expected to become masculinized if mutations are typically recessive, and feminized if they are typically dominant [15]. However, consideration of the curvature of the fitness landscape in the interval between the male and female optima has suggested that dominance coefficients will typically be reversed in comparisons of beneficial versus deleterious alleles, such that the heterozygote fitnesses are given by $(1 − h)S$ and $hT$, respectively [17,27]. This yields the conditions $(1 − h)S > hT$ for autosomal alleles, $S > 2hT$ for male-beneficial X-linked alleles, and $S > T/(2(1 − h))$ for female-beneficial X-linked alleles (note that these results are exact in the limit of weak selection; expressions for stronger selection are provided in the electronic supplementary material). Accordingly, over almost all dominance coefficients, the X chromosome promotes male-beneficial alleles over their female-beneficial counterparts [17].

How can these results be reconciled with the view that X-linked genes place greater value upon the fitness of females than that of males? The key is to take an explicitly genic,

rather than genotypic, approach. In the absence of dominance, the marginal fitness effect that a single gene has in the context of the sex in which it is advantageous is $\sigma = S/2$ if this sex is diploid at the focal locus (which is the case for both females and males if the gene is autosomal, and is the case for females if the gene is X-linked) and is $\sigma = S$ if this sex is haploid at the focal locus (which is the case for males if the gene is X-linked). Likewise, the fitness effect that the gene has in the context of the sex in which it is disadvantageous is $\tau = T/2$ if this sex is diploid at the focal locus and is $\tau = T$ if this sex is haploid at the focal locus. Accordingly, if autosomal genes place equal value on the fitness of females and males, then the condition for invasion of a mutant allele is $\sigma > \tau$, which is equivalent to $S > T$, in agreement with the above analysis. And if X-linked genes place twice the value on the fitness of females that they do males, then the condition for invasion of a mutant allele is $2\sigma > \tau$ when the allele benefits females and $\sigma > 2\tau$ when the allele benefits males, which in both cases is equivalent to $S > T$, again in agreement with the above analysis. The same logic can be used to recover the results for the dominance and reversal-of-dominance scenarios (see electronic supplementary material for details).

In other words, the X-masculinization results are entirely in line with the classic view of how X chromosomes and autosomes value female and male fitness. This equivalence has, until now, been obscured by a focus on whole genotypes and genotypic fitnesses, rather than on single genes and the fitness effects for which they—and they alone—are responsible. Specifically, X-linked genes do place an extra twofold weighting on their fitness effects in females, as a consequence of such genes spending a greater fraction of their evolutionary time in females. In this sense, X-linked genes have a female-biased agenda. However, since a gene's impact upon the phenotype may become diluted as it moves from a haploid to a diploid setting [28,29], the relative power of an X-linked gene to induce fitness effects may be lower in a female carrier than in a male. This power asymmetry creates a bias towards male-beneficial strategies that may counteract, and even overturn, the X-linked gene's more fundamental female-biased agenda.

More generally, the inclusive-fitness consequences of a gene's actions may be partitioned into three basic components: fitness effects, reproductive value and relatedness [8,30]. The fitness effects are the quantities that vary as a consequence of the gene adopting alternative strategies and represent the gene's power to shape the world. Reproductive value and relatedness together provide a currency conversion that translates these fitness effects into the gene's own inclusive-fitness valuation of any given strategy [31,32], and these dictate its agenda. The particular biological circumstances in which a gene finds itself will modulate all three components of inclusive fitness, and by investigating the modulating effects of molecular mechanisms, ecology and demography we are better able to predict and understand the relative feminization versus masculinization of sex chromosomes across different loci, populations and species (figure 1).

### (a) Fitness effects

First, we consider those factors that shape the magnitude of costs and benefits in the two sexes (figure 2). One such factor is dosage compensation. It is often assumed that the

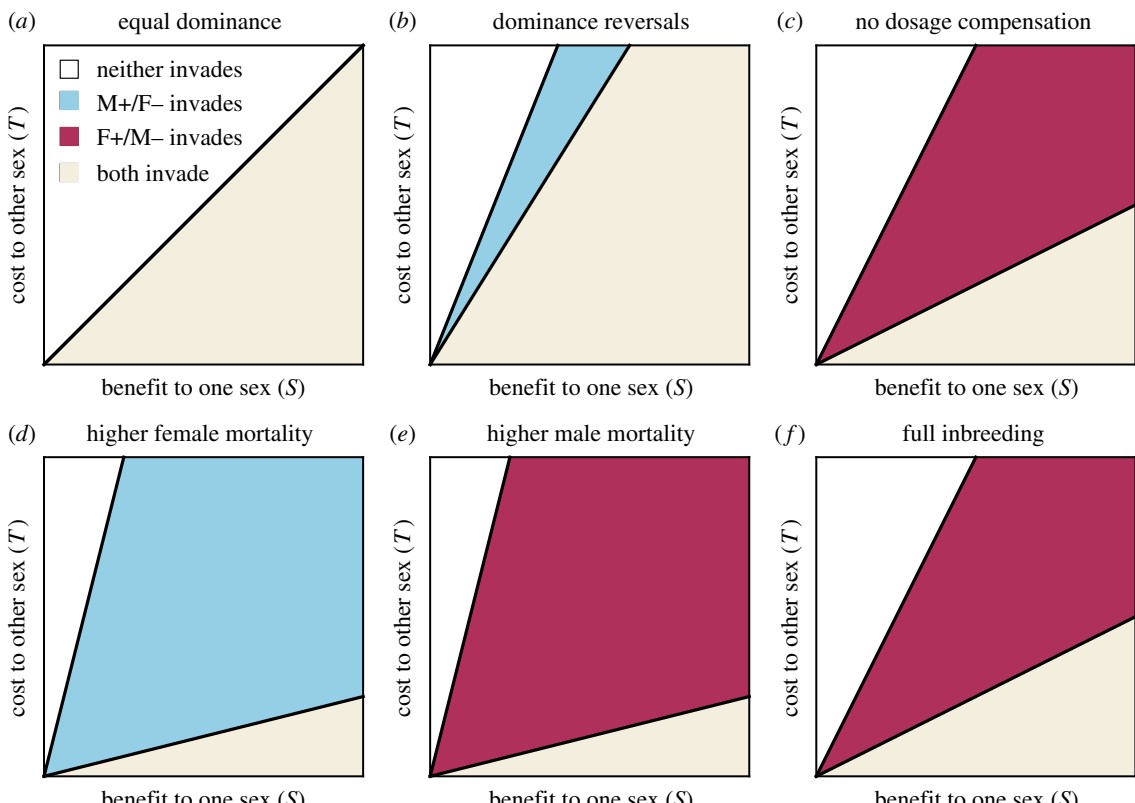

**Figure 1.** Comparison of invasion conditions for sexually antagonistic alleles on the X chromosome under weak selection. Shaded areas beneath the line indicate where an allele can invade from rarity given a whole-genotype fitness cost ($T$) to one sex and benefit to the other ($S$). ($a$) With equal dominance in the two sexes ($h = 1/2$), ($b$) with reversals of dominance between the two sexes ($h = 1/5$), ($c$) with equal dominance and no dosage compensation ($h = 1/2$, $\gamma = 0$), ($d$) with higher female mortality ($x = 3/2, y = 3$), ($e$) with higher male mortality ($x = 3, y = 3/2$), ($f$) with full inbreeding ($\phi = 1$). (Online version in colour.)

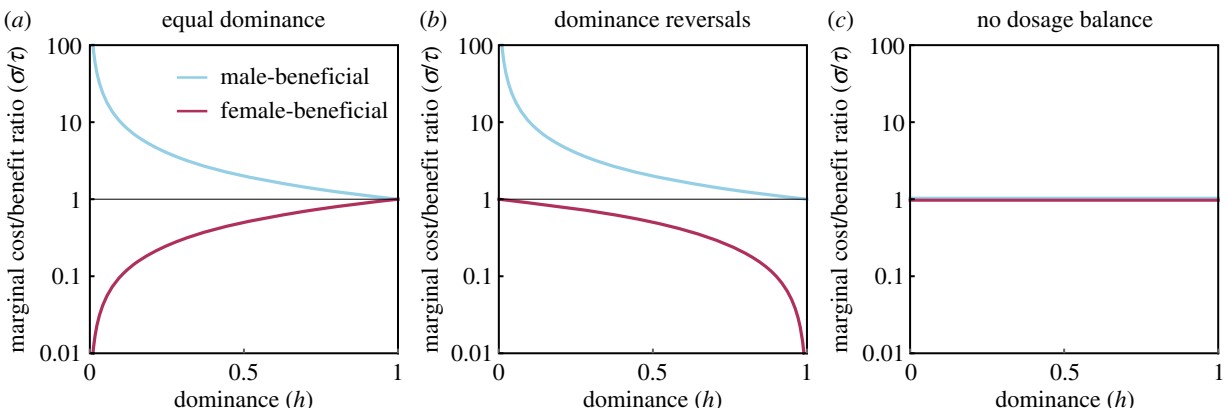

**Figure 2.** Ratios of marginal 'gene-level' fitness costs and benefits ($\sigma/\tau$) for sexually antagonistic alleles on the X chromosome under differing assumptions about dominance and dosage balance. Note the ratios are plotted on a logarithmic scale. ($a$) Equal dominance in the two sexes. ($b$) Reversals of dominance between the two sexes. ($c$) Equal dominance in the two sexes, but no dosage balance. (Online version in colour.)

phenotypes of a mutant homozygote and hemizygote are comparable [33], and consequently that the fitness effect of a single mutant X-linked allele is greater in males. This assumption is often justified by pointing to the existence of mechanisms that scale gene expression to maintain a constant X : autosome ratio of gene products across the two sexes, despite variation in the number of X chromosomes [33]. However, it is now clear that dosage-balancing mechanisms are not universal and vary across species, genes and developmental stages [34–36]. We explore the effects of this variation by introducing a parameter $\gamma$ that scales the mutant fitness effect in the heterogametic sex between the extremes of no

dosage compensation ($\gamma = 0$)—and thus comparable to the heterozygote—and full dosage compensation ($\gamma = 1$)—and thus comparable to the homozygote. Under additivity, the ratio of the marginal costs and benefits $\sigma : \tau$ in the two sexes is $(1 + \gamma)S : T$ when male-beneficial and $S : (1 + \gamma)T$ when female-beneficial. Accordingly, in the limit of full dosage compensation ($\gamma = 1$), the marginal fitness effect in males is double that in females. But, as dosage balance decreases ($\gamma < 1$), then the marginal fitness effect in males is reduced, making conditions for invasion of female-beneficial alleles less stringent and thus driving greater feminization. Other factors may also modulate the marginal fitness effects in a

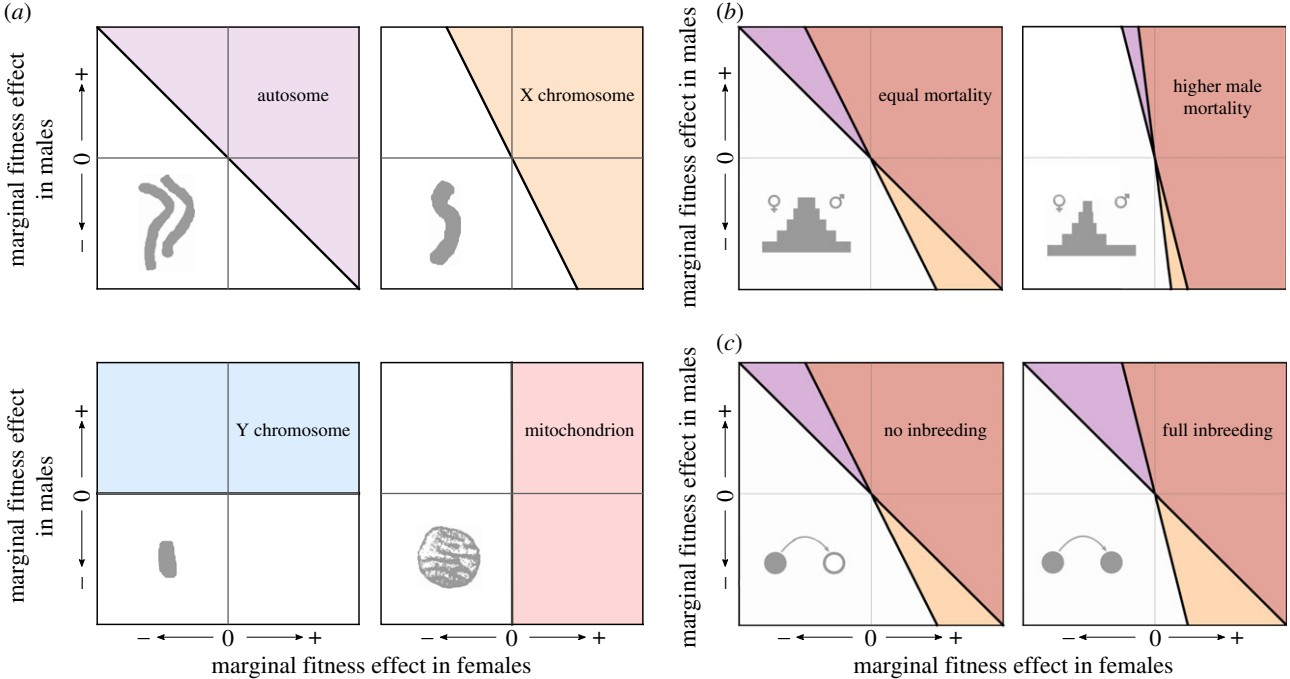

**Figure 3.** The invasion space for new mutations, illustrating the weightings put on marginal gene-level fitness effects in females and in males. (a) Across different portions of the genome (for simplicity, we assume here that mitochondria have exclusively maternal transmission), (b) for survival effects on autosomes and X chromosomes with equal mortality ($x = 2, y = 2$) and higher male mortality ($x = 3, y = 3/2$), (c) for autosomes and X chromosomes with full outbreeding ($\phi = 0$) and full inbreeding ($\phi = 1$). (Online version in colour.)

similar fashion, for instance if selection occurs predominantly in the haploid rather than diploid state of the life cycle [37], if loci are expressed in a parent-of-origin-specific manner, or if there remain functional homologues on the Y chromosome (see electronic supplementary material).

## (b) Reproductive value

Second, we consider reproductive value (figure 3). The traditional view is that a twofold weighting of female fitness effects arises because twice as many of the X-linked genes of future generations will descend from females, as compared with males, in the present generation [23,24] and, accordingly, selective effects in females are expected to shape future generations twice as strongly as are selective effects in males. However, this need not be the case in populations with overlapping generations, in which sex-biases in the stable age distribution may have a modulating effect on the reproductive values of females and males with respect to autosomal and X-chromosomal genes ([38,39]; figure 3b). Specifically, the ratio of female to male class reproductive values in an age-structured population is $x : y$ for autosomal genes and $2x : y$ for X-linked genes, where $x$ is the average age of a newborn's mother and $y$ is the average age of a newborn's father. The existence of overlapping generations means that individuals may contribute genes to the future in two ways—through survival and through reproduction—and our analysis reveals that these alternative routes are differently affected by sexual antagonism. Survival effects are weighted by the ratio of the reproductive values of female versus male survivors, which is $(x - 1) : (y - 1)$ for autosomes and $2(x - 1) : (y - 1)$ for X chromosomes. Hence, under the assumption of age-independent mortality and fecundity rates, if a sexually antagonistic X-linked allele affects survival, then it will invade when male-beneficial if $2(x - 1)\tau < (y - 1)\sigma$ and when female-beneficial if $(x - 1)\sigma > (y - 1)\tau$. By contrast, if the allele affects

fertility, then its fitness effects in males and females are valued according to their respective genetic shares of their newborn offspring. For the X chromosome, this means fertility effects are weighted in the typical 2 : 1 ratio (see electronic supplementary material).

## (c) Relatedness

Third, we consider relatedness (figure 3). Factors such as population structure and mating system may generate genetic correlations between homologous genes residing within the same individual, i.e. inbredness [40], and the traditional coefficient of inbreeding provides a measure of the relatedness between these homologues. For X-chromosomal genes, this affects males and females differently, as while females are diploid at their X-linked loci and hence can be inbred, this is not possible for males on account of their haploidy at X-linked loci [41]. As inbredness increases, we find that an X-linked gene in a female values not only its direct fitness impact on itself, but also its indirect fitness impact on the other, related, gene copy. This increases the relative importance of fitness effects in females (figure 3c). To illustrate, under a regime of assortative mating of degree $\phi$, the condition for a male-beneficial allele to invade on the X chromosome is $\sigma > 2(\tau + \phi\tau')$, and for a female-beneficial allele is $2(\sigma + \phi\sigma') > \tau$, where $\sigma'$ and $\tau'$ are the indirect fitness effects. Thus, a higher degree of assortative mating can push the invasion conditions in favour of female-beneficial alleles, even if fitness effects are of a greater magnitude in males (see electronic supplementary material).

## 3. Discussion

Taking a gene-centred approach to the problem of sexual antagonism has two major advantages. First, it provides conceptual clarity, resolving apparent contradictions between the female-biased agendas of X-chromosomal genes and the male-biased

outcomes of certain population-genetic analyses. Second, it provides a simple and practical way to separate and properly understand the factors that affect the outcome of sexually antagonistic selection. By considering in turn how different biological contexts may modulate fitness effects, reproductive values and relatedness, we can more easily generate new testable hypotheses about sexually antagonistic selection and intragenomic conflicts (some examples are given in table 1). While sexual antagonism and sex chromosome evolution have been historically well-studied topics [9,13–16], there remain significant gaps in theoretical understanding [49–52]. Here, we have shown how a gene's-eye–view approach may facilitate incorporation of salient aspects of real-world biology into future models, making them more empirically informative.

One possible avenue for future empirical investigation concerns the relationship between dosage compensation and sex-biased gene expression. While previous work has focused on how and why such dosage compensation systems may have evolved [53–56], less emphasis has been placed on how sexual antagonism may manifest differently in different dosage compensation systems (but see [42,57,58]). Given biologists' increasing knowledge of a variety of sex chromosome systems and their dosage compensation mechanisms [34–36], this presents an exciting opportunity for comparative work, both within and between species. As dosage compensation is reduced in the gonads of many species [35], we would expect greater relative feminization in gonad-expressed genes as compared with those expressed in somatic tissues. Additionally, the degree of dosage compensation may vary across sex chromosomes themselves; for example, in *Drosophila melanogaster* it is thought that the completeness of dosage compensation varies with distance from the high-affinity sites where the dosage compensation complex binds [59]. Therefore, we would expect male-beneficial alleles to invade more readily at loci close to these sites, yielding a negative relationship between male-biased gene expression and distance from these binding sites. Current evidence is mixed as to whether these new predictions are met [42–46], which may in part be due to other effects of dosage compensation upon the distribution of sex-biased genes [58]. Similar—but reversed—predictions would also apply to the Z chromosome, with increased masculinization expected for loci, tissues and species that have lower dosage compensation.

Moreover, species vary greatly in the pace and span of life [60], and within many species differences also occur between the sexes [61]. As we have shown, sex differences in life-history parameters can play an important role in shaping sexually antagonistic traits, with genes ultimately placing more value on the sex in which they spend more time. In our illustrative model, an asymmetry in mean parental age arises as a consequence of sex-specific mortality (figures 1 and 2). Thus, a novel—albeit crude—prediction would be that those organisms that typically have higher male mortality, such as mammals [48,62], will have relatively feminized genomes, while those with female-biased mortality, such as birds [63,64], will be relatively masculinzed. However, factors other than mortality may also affect mean parental age. For example, the two sexes may enter reproductive maturity at different times, and fecundity/mating success may vary with age. Consequently, one sex could have a higher mortality rate—and thus a shorter expected lifespan—yet have a higher mean parental age. An example of this is in humans, where although men typically have a higher mortality rate, the average father is older than the average mother [65,66]. This may

explain why, despite women having longer lifespans in almost all societies [67], they nonetheless senesce at a faster rate [68,69], a phenomenon that is referred to in the medical literature as the 'male–female health-survival paradox' [70,71]. While previous suggestions have been made in relation to menopause, and women's lack of direct reproduction in old age [72], the present analysis identifies the more general asymmetry in mean age of parentage in humans—whereby fathers are typically older than mothers—as a potential driver of these differences between the sexes. Additionally, for those sexually antagonistic variants affecting senescence, the later-reproducing sex would be favoured, thus further exacerbating sexual dimorphism in senescence. With demographic and genetic data on sex-specific vital rates and patterns of senescence becoming increasingly available [73–75], similar hypotheses relating intralocus sexual conflict and differences in mean parental age to sex differences in senescence and sex-biased gene expression will become testable not only in humans, but across the tree of life.

Furthermore, we have found that the asymmetry on the X chromosome between an intragenomic 'social' setting (females) and an 'asocial' one (males) means that relatedness between homologous genes may also play an important role in modulating sexual antagonism. While positive relatedness (i.e. due to inbreeding) pushes invasion conditions in favour of female-beneficial alleles, scenarios with negative relatedness (i.e. due to inbreeding avoidance) would do the opposite: with beneficial effects in females being offset by benefits to negatively related gene copies, and deleterious effects being countered by costs to negatively related gene copies (a gene-level form of spite; [22,76]). Despite the potential importance of this effect of assortative mating, few studies have explicitly considered mating scheme or population structure with regard to sexual antagonism, and those that have done so have focused instead on how these may modulate the potential for polymorphism [77–79], rather than their impact on feminization/masculinization. Specific mating systems may introduce further complications involving the relatedness of different individuals to one another—such as local mate and resource competition [80]. Although not considered here, such intrasexual and intersexual cooperative and competitive interactions can modulate the relative value of males and females [30,80], and thus potentially modulate feminization versus masculinzation of the genome. This may occur even for those genes inherited exclusively from one sex [81,82]. Combining both intra-organismal and inter-organismal social interactions provides opportunities for investigating not only how social interaction may modulate sexual antagonism but also how sexual antagonism may modulate social interaction [83,84].

Our analysis has focused mainly upon those X-linked genes for which there is no homologue on the Y chromosome, but similar principles also apply to pseudoautosomal regions. Although the dynamics of these regions are typically more complicated [85–87]—as allele frequencies may differ between males and females even if selection is weak—boundary cases are readily interpretable. We find that when recombination in males is free ($r \approx 1/2$), then these regions will evolve similarly to 'true' autosomal genes, whereas when there is no recombination—yet there remain functional copies on the Y chromosome—then X-linked genes are expected to become feminized, as while there remains the typical 2 : 1 weighting on females, the marginal fitness effects of new mutations may be expected to be of similar magnitude in males and females (see electronic supplementary material).

**Table 1.** Some empirical predictions emerging from this analysis and possible avenues for testing. n.a., not applicable.

| factor | measure of value | predictions | | | | potential empirical tests |
|---|---|---|---|---|---|---|
| | | autosomes | X chromosomes | Z chromosomes | cytoplasmic genes | |
| dosage compensation | marginal fitness effects | n.a. | increased dosage compensation → increased masculinization | increased dosage compensation → increased feminization | n.a. | • comparisons across loci with different extents of dosage compensation (e.g. distance to high-affinity sites in *Drosophila melanogaster* [42–46]) <br> • comparisons across different tissue types, (e.g. soma versus gonads of *Tribolium castanea* [47]) <br> • comparisons across different species with different degrees of dosage compensation (e.g. global versus local dosage compensation [35]) |
| mean parental age | reproductive value | • survival effects: higher mean paternal age → greater masculinization; higher mean maternal age → greater feminization <br> • fertility effects: no change | | | | • comparisons across different species with different, sex-specific mean parental ages (e.g. variation across mammals with different sex-specific mortality rates [48]) <br> • experimental evolution with direct manipulation of parental age |
| assortative mating | relatedness | no change | increased inbredness →greater feminization | increased inbredness →greater masculinization | no change | • comparisons across species with different mating structures/variation in inbredness <br> • experimental evolution manipulating the degree of inbredness |

However, if the Y chromosome degenerates, and if dosage compensation arises, then the marginal fitness effect in males will likely be larger, and thus male-beneficial alleles may more readily invade. From this, we would anticipate that X-linked alleles fixed prior to Y degeneration are more female-biased than those fixed subsequently.

Finally, while our main focus has been upon XY and XO sex determination systems, our general analysis also applies to other systems. Our results for X chromosomes can be directly applied to Z chromosomes simply by switching the roles of female and male. Similarly, the results we have obtained for autosomal regions—including those relating to age structure—will also apply to other systems with similar transmission genetics, including species that employ environmental sex determination. Along the continuum of sex-bias, the Y and W chromosomes occupy the extreme ends, as these are exclusively restricted to males and females, respectively (figure 3), and although cytoplasmically inherited genes, such as those carried by mitochondria and chloroplasts, are most often maternally transmitted, and thus expected to show extreme female bias [3], they may fall anywhere along this spectrum, depending on a combination of their mode of inheritance [88–90] and the nature of the population's age structure (see electronic supplementary material). Identifying the factors that shape the valuations these different genomic factions place on males and females—and the power they have in these different contexts—yields a richer understanding not only of the evolution of sexual dimorphism, but also of the array of intragenomic conflicts that these sex differences foment.

Data accessibility. There are no associated data. Details of the mathematical models are provided in the electronic supplementary material.

Authors' contributions. T.J.H. and A.G. jointly designed the study, performed the analysis and wrote the manuscript.

Competing interests. The authors declare that they have no conflicts of interest.

Funding. T.J.H. is supported by a PhD scholarship funded by the School of Biology, University of St Andrews. A.G. is supported by a Natural Environment Research Council Independent Research Fellowship (grant no. NE/K009524/1) and a European Research Council Consolidator (grant no. 771387).

Acknowledgements. We thank C. Barata, G. Faria, S. Frank, M. González-Forero, S. Paracchini, J. Rayner, L. Ross, L. Yusuf and two anonymous reviewers for helpful discussion.

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
