## [Reviewer comments · Proceedings of the Royal Society B: Biological Sciences]

Review History

RSPB-2020-0984.R0 (Original submission)

Review form: Reviewer 1

Recommendation

Major revision is needed (please make suggestions in comments)

Scientific importance: Is the manuscript an original and important contribution to its field?

Good

General interest: Is the paper of sufficient general interest?

Good

Quality of the paper: Is the overall quality of the paper suitable?

Good

Is the length of the paper justified?

Yes

Should the paper be seen by a specialist statistical reviewer?

Yes

Do you have any concerns about statistical analyses in this paper? If so, please specify them explicitly in your report.

No

It is a condition of publication that authors make their supporting data, code and materials available - either as supplementary material or hosted in an external repository. Please rate, if applicable, the supporting data on the following criteria.

Is it accessible?

Yes

Is it clear?

Yes

Is it adequate?

Yes

Do you have any ethical concerns with this paper?

No

Comments to the Author

In this manuscript, the authors use a modeling approach focused on the genes to resolve an apparent discrepancy regarding the evolution of sexual antagonism on the X chromosome. This model is then used in various contexts (intensity of dosage compensation, inbreeding and differential survival between sexes) to predict the evolution of sexual antagonism on the X.

The manuscript is well written and provides a theoretical background to better understand masculinization and feminization of the X chromosome depending on the genomic context (intensity of dosage compensation) and the species characteristics (inbreeding intensity and differential sex survival or age at reproduction).

It should be clarified that this model applies to cases where the Y chromosome is highly degenerated, so that the X genes are hemizygous in males. It would also be nice to say somewhere that this manuscript focuses on the XY system and a few words on the ZW system and its respective expectations would be welcome in the discussion at least.

It is unclear to me how the dosage compensation degree γ was defined. I couldn't find the information in the Supplementary material either but maybe I missed it. It seems to be a difference in expression level between the single X in males and the 2 Xs in females, is that right? This should be clarified in the text.

In the discussion lines 233-235, maybe I'm missing something, but if females reproduce more early than males, don't you expect an invasion of genes that favor males and disfavor females? In that case, how can this explain the longer longevity of females? It could explain poorer health of females but then longer longevity of females is a paradox, isn't it?

The Supplementary Materials lacks connection to the main text. Most of all, some factors are only presented in Supplementary materials (such as imprinting) and are not discussed properly.

The result section would gain in clarity by adding sub-titles on each parameter investigated.

Seeing the prediction of X feminization in the absence of dosage compensation, I was wondering if the authors found any such evidence when comparing species with and without dosage compensation using existing literature? Similarly for inbred versus outbred populations or species. Although data might not be available for such tests, it would be interesting to have a section in the discussion dedicated to listing predictions of the model and the real data required to test them.

Review form: Reviewer 2

Recommendation

Reject – article is not of sufficient interest (we will consider a transfer to another journal)

Scientific importance: Is the manuscript an original and important contribution to its field?

Marginal

General interest: Is the paper of sufficient general interest?

Marginal

Quality of the paper: Is the overall quality of the paper suitable?

Marginal

Is the length of the paper justified?

Yes

Should the paper be seen by a specialist statistical reviewer?

No

Do you have any concerns about statistical analyses in this paper? If so, please specify them explicitly in your report.

No

It is a condition of publication that authors make their supporting data, code and materials available - either as supplementary material or hosted in an external repository. Please rate, if applicable, the supporting data on the following criteria.

Is it accessible?

N/A

Is it clear?

N/A

Is it adequate?

N/A

Do you have any ethical concerns with this paper?

No

Comments to the Author

I feel this ms not well suited to PRSB. It does not derive new results regarding the evolutionary dynamics of alleles that are under sexually antagonistic selection. (These were completely worked out some time ago.) Instead, it reinterprets those dynamics in terms of concepts such as inclusive fitness, reproductive value, and a gene's "strategy". Personally, I find that this shift obscures the situation: the definitions for those terms are not always clear, and they add variables to a model that is completely defined by just three parameters (relative fitnesses). Other people, however, may find the reinterpretation a useful heuristic. If so, then this paper's value really is more philosophical than biological, and it would be more appropriate for a journal with that orientation.

Decision letter (RSPB-2020-0984.R0)

10-Jun-2020

Dear Dr Hitchcock:

I am writing to inform you that your manuscript RSPB-2020-0984 entitled "A gene's-eye view of sexual antagonism" has, in its current form, been rejected for publication in Proceedings B.

This action has been taken on the advice of referees, who have recommended that substantial revisions are necessary. With this in mind we would be happy to consider a resubmission, provided the comments of the referees are fully addressed. However please note that this is not a provisional acceptance.

Sincerely,
Dr Locke Rowe
<mailto:proceedingsb@royalsociety.org>

Editor

The reviews on this manuscript were mixed, and the rankings generally low. Referee 1, the more favourable reviewer, seeks more clarity. Referee 2 felt that there was no new fundamental result, and the conceptualization presented here may actually obscure previous results. This point makes sense to me, however, I am intrigued by the potential for new testable hypotheses. I feel that if they are there, the case needs to be made that they are inaccessible in the current framework and accessible in the proposed "gene's eye view". I also agree that it currently reads a bit more like an essay than needs be.

Reviewer(s)' Comments to Author:

Referee: 1

Comments to the Author(s)

In this manuscript, the authors use a modeling approach focused on the genes to resolve an apparent discrepancy regarding the evolution of sexual antagonism on the X chromosome. This model is then used in various contexts (intensity of dosage compensation, inbreeding and differential survival between sexes) to predict the evolution of sexual antagonism on the X.

The manuscript is well written and provides a theoretical background to better understand masculinization and feminization of the X chromosome depending on the genomic context (intensity of dosage compensation) and the species characteristics (inbreeding intensity and differential sex survival or age at reproduction).

It should be clarified that this model applies to cases where the Y chromosome is highly degenerated, so that the X genes are hemizygous in males. It would also be nice to say somewhere that this manuscript focuses on the XY system and a few words on the ZW system and its respective expectations would be welcome in the discussion at least.

it is unclear to me how the dosage compensation degree γ was defined. I couldn't find the information in the Supplementary material either but maybe I missed it. it seems to be a difference in expression level between the single X in males and the 2 Xs in females, is that right? This should be clarified in the text.

In the discussion lines 233-235, maybe I'm missing something, but if females reproduce more early than males, don't you expect an invasion of genes that favor males and disfavor females? In that case, how can this explain the longer longevity of females? It could explain poorer health of females but then longer longevity of females is a paradox, isn't it?

The Supplementary Materials lacks connection to the main text. Most of all, some factors are only presented in Supplementary materials (such as imprinting) and are not discussed properly.

The result section would gain in clarity by adding sub-titles on each parameter investigated.

Seeing the prediction of X feminization in the absence of dosage compensation, I was wondering if the authors found any such evidence when comparing species with and without dosage compensation using existing literature? Similarly for inbred versus outbred populations or species. Although data might not be available for such tests, it would be interesting to have a section in the discussion dedicated to listing predictions of the model and the real data required to test them.

Referee: 2

Comments to the Author(s)

I feel this ms not well suited to PRSB. It does not derive new results regarding the evolutionary dynamics of alleles that are under sexually antagonistic selection. (These were completely worked out some time ago.) Instead, it reinterprets those dynamics in terms of concepts such as inclusive fitness, reproductive value, and a gene's "strategy". Personally, I find that this shift obscures the situation: the definitions for those terms are not always clear, and they add variables to a model that is completely defined by just three parameters (relative fitnesses). Other people, however, may find the reinterpretation a useful heuristic. If so, then this paper's value really is more philosophical than biological, and it would be more appropriate for a journal with that orientation.

Author's Response to Decision Letter for (RSPB-2020-0984.R0)

See Appendix A.

RSPB-2020-1633.R0

Review form: Reviewer 3

Recommendation

Accept as is

Scientific importance: Is the manuscript an original and important contribution to its field?

Good

General interest: Is the paper of sufficient general interest?

Excellent

Quality of the paper: Is the overall quality of the paper suitable?

Excellent

Is the length of the paper justified?

Yes

Should the paper be seen by a specialist statistical reviewer?

No

Do you have any concerns about statistical analyses in this paper? If so, please specify them explicitly in your report.

No

It is a condition of publication that authors make their supporting data, code and materials available - either as supplementary material or hosted in an external repository. Please rate, if applicable, the supporting data on the following criteria.

Is it accessible?

N/A

Is it clear?

N/A

Is it adequate?

N/A

Do you have any ethical concerns with this paper?

No

Comments to the Author

The prior reviews clearly summarized the nature of this manuscript and the weaknesses in the previous version with regard to presentation, novelty of theory, and value of the conclusions. The authors revised the manuscript in response to each of these three broad criticisms. The revision is much stronger, particularly with regard to new predictions that follow from the novel theory.

This problem of sexual antagonism and its different consequences for various genomic components is important and timely. New technology provides opportunities for testing these ideas, with insight into the evolutionary forces that may have significantly shaped genomic interactions. I think this is a good contribution for PRSB.

I waive anonymity for transmission to the authors, but do not wish to have my name published with this review.

Decision letter (RSPB-2020-1633.R0)

20-Jul-2020

Dear Dr Hitchcock

I am pleased to inform you that your manuscript RSPB-2020-1633 entitled "A gene's-eye view of sexual antagonism" has been accepted for publication in Proceedings B.

The referee(s) have recommended publication, but also suggest some minor revisions to your manuscript. Therefore, I invite you to respond to the referee(s)' comments and revise your manuscript. Because the schedule for publication is very tight, it is a condition of publication that you submit the revised version of your manuscript within 7 days. If you do not think you will be able to meet this date please let us know.

[http://datadryad.org/submit?journalID=RSPB&manu=\(Document not available\)](http://datadryad.org/submit?journalID=RSPB&manu=(Document not available)) which will take you to your unique entry in the Dryad repository. If you have already submitted your data to dryad you can make any necessary revisions to your dataset by following the above link.

Please see <https://royalsociety.org/journals/ethics-policies/data-sharing-mining/> for more details.

Sincerely,

Dr Locke Rowe

Associate Editor

Comments to Author:

I agree with the Reviewer that the authors did a very good job at addressing the weaknesses of the manuscript and I don't have any further comments. I think this is a very nice paper that will further stimulate research in sexual conflict.

Reviewer(s)' Comments to Author:

Referee: 3

Comments to the Author(s).

The prior reviews clearly summarized the nature of this manuscript and the weaknesses in the previous version with regard to presentation, novelty of theory, and value of the conclusions. The authors revised the manuscript in response to each of these three broad criticisms. The revision is much stronger, particularly with regard to new predictions that follow from the novel theory.

This problem of sexual antagonism and its different consequences for various genomic components is important and timely. New technology provides opportunities for testing these ideas, with insight into the evolutionary forces that may have significantly shaped genomic interactions. I think this is a good contribution for PRSB.

I waive anonymity for transmission to the authors, but do not wish to have my name published with this review.

Author's Response to Decision Letter for (RSPB-2020-1633.R0)

See Appendix B.

Decision letter (RSPB-2020-1633.R1)

21-Jul-2020

Dear Dr Hitchcock

I am pleased to inform you that your manuscript entitled "A gene's-eye view of sexual antagonism" has been accepted for publication in Proceedings B.

Your article has been estimated as being 9 pages long. Our Production Office will be able to confirm the exact length at proof stage.

Open Access

Paper charges

All supplementary materials accompanying an accepted article will be treated as in their final form. They will be published alongside the paper on the journal website and posted on the online

figshare repository. Files on figshare will be made available approximately one week before the accompanying article so that the supplementary material can be attributed a unique DOI.

Sincerely,
Editor, Proceedings B
<mailto:proceedingsb@royalsociety.org>

Appendix A

Reply to Editor

Dear Dr Hitchcock:

I am writing to inform you that your manuscript RSPB-2020-0984 entitled "A gene's-eye view of sexual antagonism" has, in its current form, been rejected for publication in Proceedings B.

This action has been taken on the advice of referees, who have recommended that substantial revisions are necessary. With this in mind we would be happy to consider a resubmission, provided the comments of the referees are fully addressed. However please note that this is not a provisional acceptance.

- 1) A 'response to referees' document including details of how you have responded to the comments, and the adjustments you have made.*
- 2) A clean copy of the manuscript and one with 'tracked changes' indicating your 'response to referees' comments document.*
- 3) Line numbers in your main document.*

Sincerely,

Dr Locke Rowe

1. We are grateful for the feedback from the editors and reviewers, and for the opportunity to submit a revised version of our paper. We have fully attended to all of the comments raised, with responses provided below and with the paper revised accordingly.

In particular, we have made clearer that most of the results presented in our paper are wholly original (and not simply reinterpretations), have provided more emphasis in the main text of the results that we present in the Supplementary Material, and have spelled out more clearly the new testable predictions – and new avenues for future empirical research – that our analysis has opened up for future studies of sexual antagonism.

Reply to Associate Editor

The reviews on this manuscript were mixed, and the rankings generally low. Referee 1, the more favourable reviewer, seeks more clarity. Referee 2 felt that there was no new fundamental result, and the conceptualization presented here may actually obscure previous results. This point makes sense to me, however, I am intrigued by the potential for new testable hypotheses. I feel that if they are there, the case needs to be made that they are inaccessible in the current framework and accessible in the proposed "gene's eye view". I also agree that it currently reads a bit more like an essay than needs be.

2. This feedback has shown us that we had previously failed to make clear that the majority of the results we present in this paper are completely new and are not simply "reinterpretations" of previous findings. Indeed, our results showing a modulating effect of age structure and inbreeding on invasion conditions for sexually antagonistic alleles are *strikingly* new, and open up whole new vistas for empirical testing (as well as offering a resolution to a paradox concerning differential male/female senescence that has been raised in the medical literature).

That we also organise these new concrete results into a novel conceptual framework – synthesising inclusive-fitness theory, the gene's eye view and sexual antagonism – is, we think, a bonus, and we are disappointed that it caused one of the reviewers to write off our contribution as mere philosophy. We insist that this is a *scientific* contribution. The new framework not only helps to make sense of existing, confusing results (such as a mismatch between the classic view that the X chromosome is female-biased versus more recent theory and empirical data showing that it can be relatively masculinised), but also facilitates the development of novel testable hypotheses (as evidenced by our finding that cost/benefits are only part of the story, with relatedness and reproductive value also playing crucial roles, which has led directly to brand new testable predictions concerning assortative mating and age structure).

We have revised the paper to make the nature of our contributions clearer, including more explicitly highlighting the novelty of our results, and we have also added a Table that highlights some of our new empirical predictions and the routes by which these can be tested. This is in addition to the revisions we have made in response to specific reviewer comments – detailed below. We feel the paper is much improved, and better able to stimulate further empirical research on this exciting topic.

Referee: 1

In this manuscript, the authors use a modeling approach focused on the genes to resolve an apparent discrepancy regarding the evolution of sexual antagonism on the X chromosome. This model is then used in various contexts (intensity of dosage compensation, inbreeding and differential survival between sexes) to predict the evolution of sexual antagonism on the X.

The manuscript is well written and provides a theoretical background to better understand masculinization and feminization of the X chromosome depending on the genomic context

(intensity of dosage compensation) and the species characteristics (inbreeding intensity and differential sex survival or age at reproduction).

2. We are pleased that the reviewer enjoyed the paper, and are grateful for these kind words and this very helpful feedback.

It should be clarified that this model applies to cases where the Y chromosome is highly degenerated, so that the X genes are hemizygous in males. It would also be nice to say somewhere that this manuscript focuses on the XY system and a few words on the ZW system and its respective expectations would be welcome in the discussion at least.

3. Yes, our main focus has been on X-linked genes for which there are no homologues on the Y chromosome. We now make this clear, and provide a thorough discussion of the situation for pseudoautosomal regions (for which homologues do exist), including how the balance of feminization versus masculinisation is expected to shift over the course of Y degeneration (lines 70-71, 162-163 and 290-303). We also spell out how our results also apply to ZW systems, by reversing the roles of females and males – and also to environmental sex determination (lines 235-237, 305-310, plus entries in the new Table 1).

it is unclear to me how the dosage compensation degree γ was defined. I couldn't find the information in the Supplementary material either but maybe I missed it. it seems to be a difference in expression level between the single X in males and the 2 Xs in females, is that right? This should be clarified in the text.

4. The idea here is that, *a priori*, it's not obvious whether an individual's phenotype should depend on the number of copies they carry of a particular allele or whether it is the frequency (concentration) of that allele in their genome that matters. If it is copy number *per se*, then the hemizygote should be comparable with the heterozygote (as they both carry one copy of the allele); and if it is frequency *per se*, then the hemizygote should be comparable with the homozygote (as they both carry the allele at frequency 1). The parameter γ scales between these two scenarios (the former corresponds to $\gamma = 0$, the latter corresponds to $\gamma = 1$) – and also allows for everything in-between. Full dosage compensation implies that the hemizygote is comparable to the homozygote ($\gamma = 1$), and the absence of dosage compensation implies that the hemizygote is comparable to the heterozygote ($\gamma = 0$). We spell this out more clearly in the revised paper (lines 150-157).

In the discussion lines 233-235, maybe I'm missing something, but if females reproduce more early than males, don't you expect an invasion of genes that favor males and disfavor females? In that case, how can this explain the longer longevity of females? It could explain poorer health of females but then longer longevity of females is a paradox, isn't it?

5. That's exactly the point. In the medical literature, the empirical observation that women tend to live longer than men, but tend to senesce faster than men, has been described as a paradox. And our analysis yields a potential resolution to this paradox: men appear to die at a younger age than women (on average) but reproduce at an older age (in terms of a newborn's father being older, on average, than the mother), and our analysis suggests that this situation would lead to women exhibiting a greater rate of senescence

despite living longer lives. We've now provided an expanded discussion to make the logic clearer (lines 242-261).

The Supplementary Materials lacks connection to the main text. Most of all, some factors are only presented in Supplementary materials (such as imprinting) and are not discussed properly.

6. We have made more references to the Supplementary Material, including adding brief discussion of the novel results provided therein, throughout the main text. This makes clearer that our analysis also covers haploid selection, genomic imprinting, pseudoautosomal regions, and cytoplasmic genes (lines 159-163;290-303;305-320 and the new Table 1). We have also introduced a summary section into the Supplementary Material to make cross-referencing between this and the main text more straightforward.

The result section would gain in clarity by adding sub-titles on each parameter investigated.

7. Done.

Seeing the prediction of X feminization in the absence of dosage compensation, I was wondering if the authors found any such evidence when comparing species with and without dosage compensation using existing literature? Similarly for inbred versus outbred populations or species. Although data might not be available for such tests, it would be interesting to have a section in the discussion dedicated to listing predictions of the model and the real data required to test them.

8. Owing to the novelty of the issues that we are exploring in our paper, in some instances the kinds of data that would be needed to test our predictions have simply not been provided by previous studies. But where we have found relevant data – such as on dosage compensation – we have included it. Our intention is that our analysis will spur new empirical research activity on this topic, and in order to facilitate this we have now provided further discussion of how our predictions can be tested and have highlighted caveats concerning possible confounding variables (lines 244-249, 279-285 and the new Table 1).

Referee: 2

I feel this ms not well suited to PRSB. It does not derive new results regarding the evolutionary dynamics of alleles that are under sexually antagonistic selection. (These were completely worked out some time ago.) Instead, it reinterprets those dynamics in terms of concepts such as inclusive fitness, reproductive value, and a gene's "strategy". Personally, I find that this shift obscures the situation: the definitions for those terms are not always clear, and they add variables to a model that is completely defined by just three parameters (relative fitnesses). Other people, however, may find the reinterpretation a useful heuristic. If so, then this paper's value really is more philosophical than biological, and it would be more appropriate for a journal with that orientation.

9. We have made clearer in the revised paper that most of our results are completely novel and are not simply reinterpretations of existing work (see above). For example, our predictions concerning how sex-specific age structure modulates the balance between female-beneficial versus male-beneficial alleles is strikingly new, lead to completely novel empirical predictions for sexual antagonism that have not previously been anticipated in the evolutionary biology literature, and offer a resolution to the paradox of female versus male rates of senescence that has been raised in the medical literature. We have more carefully spelled out the new avenues of empirical testing that our predictions have opened up, to spur future advances on this topic.

We find it surprising that the reviewer feels that the evolutionary dynamics of sexually antagonistic alleles “were completely worked out some time ago”, given the repeated, recent calls for new theory on exactly this problem (e.g. Abbot et al. 2017, Furman et al. 2020, both cited in our paper) and other recent modelling that has been undertaken on this topic (e.g. Harts et al. 2014, Connallon et al. 2019, Patten 2019 all cited in our paper). We have given these recent articles more prominence in our paper.

Our new organisational framework – which emerges as a synthesis of the theory of inclusive fitness, the gene’s eye view of adaptation, and the problem of sexual antagonism – has been central to motivating and deriving these novel results. The “three measures of value” outlook, borrowed from the theory of inclusive fitness, served to directly draw our attention to possibilities that have not previously been anticipated: in particular, the reproductive value consequences of sex-specific age structure, and the relatedness consequences of assortative mating. We anticipate that other researchers will also find this approach useful in developing their own novel, testable predictions. By facilitating a tight interplay of theoretical and empirical research, this framework constitutes a scientific – rather than a philosophical – contribution.

Inclusive fitness theory, the gene’s eye view, and sexual antagonism have all been of huge interest to evolutionary biologists for decades, and continue to have direct relevance to scientific researchers working across other domains, from anthropology to molecular biology to medicine. So we believe there is an audience for this research at *Proceedings B*.

Appendix B

(Responses in bold.)

Dear Dr Hitchcock

I am pleased to inform you that your manuscript RSPB-2020-1633 entitled "A gene's-eye view of sexual antagonism" has been accepted for publication in Proceedings B.

The referee(s) have recommended publication, but also suggest some minor revisions to your manuscript. Therefore, I invite you to respond to the referee(s)' comments and revise your manuscript. Because the schedule for publication is very tight, it is a condition of publication that you submit the revised version of your manuscript within 7 days. If you do not think you will be able to meet this date please let us know.

Online supplementary material will also carry the title and description provided during submission, so please ensure these are accurate and informative. Note that the Royal Society will not edit or typeset supplementary material and it will be hosted as provided. Please ensure that the supplementary material includes the paper details (authors, title,

journal name, article DOI). Your article DOI will be 10.1098/rspb.[paper ID in form xxxx.xxxx e.g. 10.1098/rspb.2016.0049].

If you wish to submit your data to Dryad (<http://datadryad.org/>) and have not already done so you can submit your data via this

link [http://datadryad.org/submit?journalID=RSPB&manu=\(Document](http://datadryad.org/submit?journalID=RSPB&manu=(Document) not available) which will take you to your unique entry in the Dryad repository. If you have already submitted your data to dryad you can make any necessary revisions to your dataset by following the above link.

Please see <https://royalsociety.org/journals/ethics-policies/data-sharing-mining/> for more details.

Sincerely,

Dr Locke Rowe

Associate Editor

Comments to Author:

I agree with the Reviewer that the authors did a very good job at addressing the weaknesses of the manuscript and I don't have any further comments. I think this is a very nice paper that will further stimulate research in sexual conflict.

Response to editor

1. We are grateful for the kind feedback from the editor, and are glad that we have managed to address the weaknesses of the previous submission. We have made some minor changes to the MS, including some minor corrections to Figure 1 and Table 1, as well as introducing an explicit reference to Table 1 in the main text (line 212).

Reviewer(s)' Comments to Author:

Referee: 3

Comments to the Author(s).

The prior reviews clearly summarized the nature of this manuscript and the weaknesses in the previous version with regard to presentation, novelty of theory, and value of the conclusions. The authors revised the manuscript in response to each of these three broad criticisms. The revision is much stronger, particularly with regard to new predictions that follow from the novel theory.

This problem of sexual antagonism and its different consequences for various genomic components is important and timely. New technology provides opportunities for testing these ideas, with insight into the evolutionary forces that may have significantly shaped genomic interactions. I think this is a good contribution for PRSB.

2. We thank the reviewer for their kind words. We are pleased that they feel that our revisions have improved the paper, drawing out the novelty and the empirical predictions.